# Lyme-Borreliosis Disease: IgM Epitope Mapping and Evaluation of a Serological Assay Based on Immunodominant Bi-Specific Peptides

**DOI:** 10.3390/biomedicines13081930

**Published:** 2025-08-08

**Authors:** Mônica E. T. A. Chino, Paloma Napoleão-Pêgo, Virgínia L. N. Bonoldi, Gilberto S. Gazeta, João P. R. S. Carvalho, Carlos M. Morel, David W. Provance-Jr, Salvatore G. De-Simone

**Affiliations:** 1Center for Technological Development in Health (CDTS)/National Institute of Science and Technology for Innovation in Neglected Population Diseases (INCT-IDPN), Oswaldo Cruz Foundation (FIOCRUZ), Rio de Janeiro 21040-900, RJ, Brazil; elizaalcon@gmail.com (M.E.T.A.C.); pegopn@fiocruz.br (P.N.-P.); joaopedrorsc@gmail.com (J.P.R.S.C.); cmmorel@fiocruz.br (C.M.M.); bill.provance@fiocruz.br (D.W.P.-J.); 2Post-Graduation Program in Science and Biotechnology, Department of Molecular and Cellular Biology, Biology Institute, Federal Fluminense University, Niteroi 22040-036, RJ, Brazil; 3Clinical Hospital, Faculty of Medicine, São Paulo University, São Paulo 05403-000, SP, Brazil; virginia.bonoldi@hc.fm.usp.br; 4Laboratory of Ticks and Other Wingless Arthropods/National Reference for Vectors of Rickettsioses, Instituto Oswaldo Cruz-IOC, Oswaldo Cruz Foundation (FIOCRUZ), Rio de Janeiro 21040-900, RJ, Brazil; gsgazeta@ioc.fiocruz.br; 5Program of Post-Graduation on Parasitic Biology, Oswaldo Cruz Institute, Oswaldo Cruz Foundation (FIOCRUZ), Rio de Janeiro 21040-900, RJ, Brazil

**Keywords:** Lyme-borreliosis, IgM epitopes, serodiagnosis, peptides, ELISA

## Abstract

Lyme borreliosis (LB) is a tick-borne infection of global relevance that remains underrecognized, hindering effective surveillance and diagnosis. This lack of awareness and the limited specificity and low antibody titters of current serological assays underscore the need for improved diagnostic tools. Here, we investigated the molecular fine specificity of IgM antibody responses to five proteins of *Borrelia burgdorferi*. **Materials and Methods:** We employed peptide arrays on cellulose support (SPOT synthesis) to screen IgM epitopes and assess cross-reactivity through databank searches and Enzyme-Linked Immunosorbent Assay (ELISA). Validation was performed using ELISA and Receiver Operating Characteristic (ROC) curve analysis. **Results:** We identified ten IgM epitopes, of which four were classified as specific. The ELISA peptide assay demonstrated a sensitivity of ≥87.3%, specificity of ≥56.2%, and accuracy of ≥66.6%. A bi-specific peptide was subsequently synthesized and evaluated by ELISA using a panel of patient sera representing different pathologies. This result showed a sensitivity of 85.0% and a specificity of 100.0%, with significant differences in cross-reactivity between the leptospirosis and syphilis groups. **Conclusions:** These findings indicate that the identified peptide combinations could facilitate the development of new, highly specific serodiagnostic assays, thereby enhancing public health initiatives and epidemiological studies.

## 1. Introduction

Lyme borreliosis (LBD) is a zoonotic disease caused by the *Borrelia burgdorferi* sensu lato complex spirochetes and transmitted by ticks of the genus *Ixodes*. While predominantly described in the Northern Hemisphere, a similar clinical presentation has been reported in Brazil under the designation of Baggio–Yoshinari Syndrome (SBY), which is considered a regional variant of Lyme-like borreliosis [1,2,3,4,5,6].

Recently, the discovery of multiple *Borrelia* genospecies with distinct genomes and antigenic profiles has highlighted the group’s genetic and immunological diversity, impacting clinical presentation and diagnostic accuracy [7,8,9,10,11,12]. Consequently, these facts pose enormous difficulties in diagnosis and complexity in clinical management, especially in advanced stages.

LB is a multisystemic and treatable disease when diagnosed early. However, its broad clinical spectrum, overlapping with other infectious and autoimmune conditions, and the complexity of its immune response hinder early and accurate diagnosis [13,14,15,16]. Standard laboratory testing consists of a two-tier serological algorithm: a screening ELISA and Western blot confirmation. Although widely used, this approach is limited by low sensitivity during the early phase of infection, potential cross-reactivity with other pathogens, and poor applicability in regions where circulating strains differ from those used to generate the diagnostic antigens [17,18].

To overcome these limitations, recombinant proteins and synthetic peptides derived from well-characterized epitopes have been proposed. These subunit antigens offer greater purity, consistency, and specificity than crude extracts, minimizing cross-reactivity [19,20,21,22,23]. Among these antigens, outer surface proteins (Osp) and flagellar hook proteins (FlgE, FliD) play critical roles in the pathogenesis and immune evasion of *Borrelia* and are targets of strong humoral responses, especially during early infection [24,25,26,27,28,29,30,31]. Although IgG is classically used to assess past or chronic infections, IgM antibodies are produced during the early stages of infection and serve as critical markers for acute-phase diagnosis [30,31,32,33]. Following antigen stimulation, class switching increases antibody affinity without changing specificity [34]. Persistent or recurrent IgM responses have been observed in chronic and post-treatment LB, further supporting the relevance of targeting IgM epitopes [34,35,36].

As a result, antibodies generated by infection with *B. burgdorferi* have been of great interest in confirming infection. Serological diagnosis of LB disease based on CDC criteria has been used [37]; however, in Brazil, it can be problematic and lead to false-negative diagnoses [38,39]. In addition to detecting antibodies against acute and chronic infections, serological tests can be used for interventions with antimicrobials and to understand the epidemiology and surveillance of LB-like syndrome, which can guide public health, given that it is a zoonotic disease [40]. Other rapid serological tests have been developed, and epitope-specific antibody serological assays have improved specificity [40,41,42,43].

Conversely, high-throughput biomarker analysis methods have been developed to identify disease risk and diagnostic, prognostic, and therapeutic targets in human clinical samples [44]. Protein microarrays are perfectly suited to fill the gap between biomarker discovery and diagnostics on the same platforms. Disease signatures can be identified, validated, and used for routine diagnostics.

In contrast to refining existing diagnostic tools, identifying novel IgM epitopes offers a more targeted strategy to improve specificity in the early phase of infection, when serological markers are typically variable. This approach may circumvent current limitations by focusing on highly immunoreactive regions that elicit robust early antibody responses.

Thus, this study aims to elucidate the IgM epitopes of five flagellar and outer surface proteins of *B. burgdorferi*, considering Baggio–Yoshinari Syndrome as a clinical expression of Lyme-like borreliosis in Brazil. Our findings may support the development of accurate, early-phase serological assays based on conserved epitopes relevant to regional strains.

## 2. Materials and Methods

### 2.1. Human Serum Samples

A total of 82 human serum samples were analyzed and classified into three groups: (i) seropositive cases (*n* = 19), confirmed by ELISA and Western blot, clinical features and epidemiological details; (ii) suspected cases (*n* = 24), presenting clinical features compatible with Lyme disease but negative in serological assays; and (iii) seronegative controls (*n* = 39), consisting of healthy blood donors with no clinical or epidemiological evidence of Lyme disease.

The 24 sera from suspected cases were obtained from patients clinically diagnosed with Lyme disease-like syndrome. Among these, 11 samples were collected more than three months after symptom onset and 13 within the first three months (Table 1). Seropositive and suspected case samples were provided by the Rheumatology Division at the University of São Paulo School of Medicine (USP) and the Laboratory of Biodiversity in Entomology at the Oswaldo Cruz Institute (IOC/FIOCRUZ). Seronegative samples were obtained from the Hemotherapy Institute of Rio de Janeiro (HEMORIO).

The Ethics in Research Committee (CEP-25836019.0.0000.5243)—UFF/FIOCRUZ approved the experiments involving human sera samples. All samples were obtained from external laboratories and collected under their respective ethical approvals, including informed consent from participants.

### 2.2. Preparation of the Cellulose Membrane-Bound Peptide Array

The complete sequence of *B. burgdorferi* (Q44767, P11089, O51173, Q44849, and P0CL66) was covered by synthesizing 15-residue peptides with a 10-residue overlap, automatically prepared on cellulose membranes (Amino-PEG 500-UC540) following the standard SPOT synthesis protocol using an Auto-Spot Robot ASP-222 (Intavis Bioanalytical Instruments AG, Köln, Germany) [45]. Positive control peptides, including [IHLVNNESSEVIVHK and GYPKDGNAFNNLDRI] (*Clostridium tetani*, spots P20 and P21), KEVPALTAVETGATN (*Poliovirus*, spot P22), and YPYDVPDYAGYPYDV (*H. influenza* virus hemagglutinin, spot P23), were included, and programming was conducted using Multipep software (Intavis Bioanalytical Instruments AG, Köln, Germany). The library contained 371 peptides plus four positive control peptides (Appendix A). Coupling reactions were followed by acetylation with 4% acetic anhydride in N, N-dimethyl formamide to make the peptides N-reactive for subsequent steps. After acetylation, F-moc protecting groups were removed with piperidine to activate the peptides. Additional amino acids were added through coupling, blocking, and deprotection until the desired peptide was obtained. Once the final amino acid was added, sidechain deprotection was performed using a solution of dichloromethane-trifluoracetic acid-triisopropylsilane (1:1:0.05, *v*/*v*/*v*), followed by ethanol washing as described previously [46]. The membranes containing the synthetic peptides were immediately probed.

### 2.3. Evaluation of SPOT Membranes

SPOT membranes were washed for 10 min with TBS-T buffer (50 mM Tris, 136 mM NaCl, 2 mM KCl, and 0.05% Tween, pH 7.4) and then blocked with TBS-T containing 1.5% BSA for 90 min at 8 °C with agitation. After extensive washing with TBS-T, the membranes were incubated for 12 h with a pool (*n* = 7) of Lyme disease patient sera (1:150) in TBS-T + 0.75% BSA, followed by another wash with TBS-T. The membranes were then incubated with goat anti-human IgM (µ-chain specific) conjugated with alkaline phosphatase (anti-human IgM, 1:5000; Sigma Chem, St Louis, MO, USA) for 1 h and washed with TBS-T and CBS (50 mM citrate-buffered saline). Finally, chemiluminescent CDP-Star^®^ substrate (Cytiva, Marlborough, MA, USA) (0.25 mM) with Nitro-Bloc-II™ Enhancer (Cat # T2218, Applied Biosystems, Waltham, MA, USA) was added to complete the reaction for 5 min.

### 2.4. Scanning and Quantification of Spot Signal Intensities

Chemiluminescent signals were detected using an Odyssey FC (LI-COR Bioscience, Lincoln, NE, USA) under previously established conditions [47], with minor modifications. In brief, a digital image file with a 5 MP resolution was generated, and signal intensities were quantified using TotalLab TL100 software (v 2009, Nonlinear Dynamics, Newcastle Upon Tyne, UK). This software includes an automatic grid search for 384 spots but does not identify potential epitope sequences automatically. Therefore, the data were further analyzed using Microsoft Excel. For a sequence to be considered an epitope, two or more positive contiguous spots must have a signal intensity (SI) equal to or greater than 30% of the highest value from the set of spots on the corresponding membrane. Signal intensity (SI) for the background was defined by a set of negative controls on each membrane.

### 2.5. Peptide Preparation

The selected individual peptides were synthesized via the solid-phase chemical method using the 9-fluorenyl methoxycarbonyl (F-moc) strategy on an automated peptide synthesizer (MultiPep-1, CEM Corp, Charlotte, NC, USA), as previously described [48]. Peptide concentrations were determined by measuring optical density and applying the molar extinction coefficient calculated by the PROTPARAM software package [http://www.expasy.ch; accessed on 15 July 2023]. The peptide sequences were confirmed using mass spectrometry (MALDI-TOF MS) (Matrix-Assisted Laser Desorption Ionization Time-of-Flight).

### 2.6. Synthesis of Bi-Specific Antigen Peptides

The individual epitope peptides Bburg/02/huG and Bburg/06/huG were synthesized in tandem, incorporating a GGGG interpeptide spacer. For the preparation of the multi-antigen peptide (MAPs4), a polyethylene glycol grafted TentaGel^®^M NH2 resin (RAPP Polymer, Tübingen, Land Baden-Württemberg, Germany) and the standard solid-phase synthesis protocol was used. The constructs were synthesized on an automated peptide synthesizer (MultiPep1, CEM Corp, Charlotte, NC, USA), following the established method [45]. HPLC-purified peptides and their identities were confirmed by MS (MALDI-TOF or electrospray).

### 2.7. Enzyme-Linked Immunosorbent Assay (ELISA)

The ELISA was conducted as previously outlined, with slight adjustments [47]. The analysis used the same antigen concentration for synthetic single peptides and the bi-peptide. Synthetic peptides (0.5 µg/well) were immobilized on Immulon 4HB flat-bottom 96-well microtiter plates (Corning, Corning, NY, USA) by coating with 100 µL of each peptide dissolved in coating buffer (0.1 M sodium carbonate-bicarbonate, pH 9.6) and incubated overnight at 4 °C. Following each incubation step, the plates (Immulon 4HBx) were washed three times with PBS-T (phosphate-buffered saline with 0.1% Tween 20, pH 7.2) and blocked with 200 µL of PBS-T containing 2.5% BSA for 2 h at 37 °C.

Subsequently, the plates were incubated for 1 h at 37 °C with 50 µL of serum from patients with Lyme disease-like syndrome, diluted in the blocking buffer (1:100 for single peptides and 1:50 for polypeptides). After additional washes, the plates were treated with 100 µL of goat anti-human IgM (μ-chain specific)—HRP (1:20,000, Sigma-Aldrich, St Louis, MO, USA) for 1 h at 37 °C. The reaction was developed using Chemiluminescent CDP-Star^®^ Substrate with Nitro-Block-II™ Enhancer (Applied Biosystems, Waltham, MA, USA) substrate for 15 min, and absorbance at 405 nm was recorded with a Hidex Sense Microplate Reader (Turku, Finland). A response was deemed elevated if the optical density exceeded the threshold, defined as the mean of the negative controls plus three standard deviations.

### 2.8. Computational Tools

The protein sequences of interest (Q44767, P11089, O51173, Q44849, and P0CL66) from the *B. burgdorferi* American strain were initially retrieved from the UniProt database [http://www.uniprot.org/] (accessed on 11 August 2022). Similarity analysis with proteins from other organisms was conducted using BLASTP version 4.0. To identify the epitope locations within the 3D structures of these proteins, we generated models using UCSF ChimeraX, Version 1.5 (https://www.rbvi.ucsf.edu/chimerax/; accessed on 12 August 2021). Structure generation of the five proteins was performed by employing the AlphaFold server [49] and the Protein Data Bank (PDB). Searches for *B. burgdorferi* peptides were conducted in the Protein Information Resource (PIR) database (https://research.bioinformatics.udel.edu/peptidematch/index.jsp, accessed on 12 April 2022) using previously identified sequences in other organisms.

### 2.9. Data Analysis

The ELISA test results were statistically analyzed using MedCalc software version 23.2.0 (https://apps.microsoft.com/detail/9nlfqd28xtnp?hl=pt-BR&gl=BR, accessed on 12 April 2023). The OD data were exported to GraphPad Prism software version 10, and the cutoff values were defined based on the ROC curve analysis. Each peptide’s reactivity index (RI) was calculated as the ratio of the optical density (OD) of a specific sample to the cutoff OD values for each test. Sensitivity, specificity, and cross-reactivity with leptospirosis and syphilis samples were also assessed. RI values were classified as positive (>1.00) or negative (≤1.00), with a gray zone defined as RI values within ±10% of the cutoff, where results may be ambiguous. Statistical analysis of cross-reactivity was performed using GraphPad Prism version 5.0. Differences between groups were assessed using the Kruskal–Wallis test, with statistical significance set at a *p*-value ≤ 0.05. A one-way ANOVA with the Kruskal–Wallis post-test was used to compare multiple groups, with *p*-values < 0.005 considered statistically significant

## 3. Result

### 3.1. Detection of IgM Epitopes in Surface Proteins of B. burgdorferi

Proteins are known to contain a finite number of epitopes that are recognized by antibodies, but their molecular interactions remain largely undefined. In the case of Borreliosis, only a few proteins have been extensively studied. Therefore, this study analyzed five membrane proteins by synthesizing a library of representative peptides. These peptides were incubated with human seropositive samples and IgM secondary antibodies (see Section 2). As shown in Figure 1A, the analysis of signal intensity (with a cutoff above 30%) of the primary sequences from the proteins containing B cell epitopes in the 384-peptide library (Figure 1B and Appendix A) identified 14 peptides that were reactive to IgM from a pool of patients’ sera. Table 2 summarizes the immunoreactivities of the identified peptides, which include two peptides from the flagellar hook protein (Bburg/01/huM and Bburg/02/huM), two peptides from the 41-kDa flagellin core protein (Bburg/03/huM and Bburg/04/huM), four peptides from the flagellar hook-associated protein 2 (Bburg/06/huM to Bburg/09/huM), two peptides from the putative outer membrane protein BBA03 (Bburg/10/huM and Bburg/11/huM), and two peptides from outer surface protein A (Bburg/12/huM and Bburg/13/huM).

### 3.2. Secondary Structure and Structural Mapping of the IgM Epitopes

The I-TASSER service predicted the protein’s secondary structure components in detail. Molecular modeling generated five protein models, ranked according to their C-scores and TM-scores, with the lowest values appearing first. The models with the highest C-scores and TM-scores were chosen based on their secondary structures. Table 2 illustrates the secondary structures of the 14 peptides containing epitopes ranging from 4 to 14 amino acids and their positions within each protein. These peptides exhibited reactive epitopes in their secondary structures’ coil, helix, and strand regions. PIR alignment showed that four peptides exhibited similarities to peptide sequences from various *Borrelia* species, while the remaining peptides shared homology with sequences from other animals (Table 2).

The spatial localization of linear B-cell IgM epitopes was identified and mapped onto 3D protein structures, as illustrated in Figure 2. Figure 2A,B represents flagellar proteins, while Figure 2C displays a surface protein. Red-circled epitopes were selected for validation, highlighting their exposed surface positioning, which enhances accessibility for antibody binding. The spatial arrangement of the coil within the outer membrane protein was examined via 3D structural analysis using ChimeraX, which revealed that these epitopes reside within the coiled-coil region of the flagellar protein, located between domain 1 (D1) and domain 2 (D2) and extend near the N-terminal.

Figure 2D shows the surface cellulose presenting the sequence and increasing signal intensity, providing a detailed visual representation of the reactivity with IgM antibodies and correlating with greater serum levels of the target, suggesting differences in epitope immunogenicity.

### 3.3. ELISA Screening

The reactivity index (IR) and gray zone (GZ) were obtained by ELISA using the synthetic peptides Bburg/01/huM, Bburg/02/huM, Bburg/06/huM and Bburg/13/huM. The result obtained is shown in Figure 3. These four IgM-reactive peptides were selected based on signal intensity, predicted secondary structure, and peptide match. The peptide-ELISA was evaluated on a panel of 19 seropositive, 39 seronegative, and 24 sera from patients with suspected Lyme disease. Among the suspected cases, 11 samples were collected more than three months after symptom onset and 13 within three months. The grey zone (GZ) to RI value was 1.0 ± 10%.

Table 3 presents the performance of peptides in an ELISA assay with samples from patients with Borreliosis. The cutoff was determined using the ROC curve, and the peptides (Bburg/02/huM and Bburg/06/huM) demonstrated a potential capacity to detect positive cases with a sensitivity of ≥94.74%. Specificity was ≥97.44%, as seen in Table 3, suggesting strong specificity, which means they could properly identify negative cases.

On the other hand, peptides (Bburg/01/huM and Bburg/13/huM) showed an antagonistic profile concerning sensitivity and specificity. According to Table 3, peptide Bburg/01/huM presented high sensitivity. However, the specificity was slightly lower, equivalent to 56.2%. The performance of peptide Bburg/13/huM revealed a sensitivity of 47.37%, suggesting a relatively low capacity to identify the infected individuals. However, the specificity, which is the ability to identify the non-infected, was high.

Furthermore, the (Area under the curve) AUC was determined for each peptide, and the findings showed that peptides Bburg/02/huM and Bburg/06/huM 162 were highly accurate. The accuracy of peptide Bburg/01/huM was good, suggesting a good performance in discriminating between infected and non-infected patients. Peptide Bburg/13/huM, on the other hand, had a lower AUC value, indicating a less accurate performance. According to the analysis, the peptides were evaluated and found to have a significant discriminatory ability with a *p*-value or significance level of (*p* < 0.0001) to peptides (Bburg/02/huM and Bburg/06/huM), Bburg/01/huM (*p* = 0.0052), and Bburg/13/huM (*p* = 0.0703).

We saw that the peptides Bburg/02/huM and Bburg/06/huM used in the ELISA assays with samples from suspected cases exhibited a sensitivity of ≥ 92.3% and a specificity of 100.0% (Table 3). In contrast, Bburg/01/huM and Bburg/13/huM exhibited sensitivities of 72.7% and 56.2%, respectively, with no statistically significant variation. Furthermore, the number of potentially laboratory-confirmed cases (defined by a Reactivity Index > 1.0 in peptide-based ELISA) was as follows: Bburg/01/huM (16/24), Bburg/02/huM (23/24), Bburg/06/huM (23/24), and Bburg/13/huM (6/24).

### 3.4. Potential Cross-Reactivity of the Epitopes In Silico

The BLASTP analysis of *B. burgdorferi* proteins [Flagellar E (FlgE, Q44849) and outer surface protein A (OspA, P0CL66)] examined in this study revealed a significant alignment with proteins from other *Borrelia* species, indicating that the identified *Borrelia*-specific epitopes are universal. In contrast, the flagellar filament 41 kDa protein (Flg 41 kDa, Q4767), flagellar hook-associated protein 22 (Flg hook 2, PI1089), and Omp BBA03 (Omp BBA03, O51173) showed sequence similarities ranging from 39.29% to 57.4% with *Treponema pallidum* and *Leptospira interrogans* (Appendix A).

### 3.5. Potential Cross-Reactivity and Validation of Bi-Specific Peptides as Antigens

When the same patient’s sera were tested against anti-LD IgM to validate the bi-specific peptide as an antigen, the ELISA results demonstrated a sensitivity of 85.0% and a specificity of 100.0%, comparable to those obtained with single peptides, with a statistically significant *p*-value of 0.001 (Figure 4).

An ELISA assay was employed to evaluate cross-reactivity using serum samples from patients with Lyme-borreliosis (*n* = 20), leptospirosis (*n* = 20), and syphilis (*n* = 20). The synthetic bi-peptide ELISA assay revealed that only 2 out of 20 syphilis samples and 5 out of 20 leptospirosis samples were reactive for IgM antibodies. In contrast, Dunn’s multiple comparisons test, conducted as a post-hoc analysis to pinpoint specific pairwise differences, showed the LB group exhibited a considerably higher median reactivity (0.2613) compared to leptospirosis (0.0755) and syphilis (0.02175) (Table 4).

Statistical analysis with the Kruskal–Wallis test indicated significant differences between the groups (*p* < 0.0001). Furthermore, pairwise comparisons using the LB group as a reference confirmed that the differences in reactivity were statistically significant for both leptospirosis (*p* < 0.0424) and syphilis (*p* < 0.0001) (Figure 5).

Despite the small number of reactive samples in the leptospirosis and syphilis groups, these results demonstrate that the bi-peptide reactivity is significantly higher in LB patients, underscoring the assay’s specificity for this condition.

## 4. Discussion

In this study, we investigate the antigenic properties of five surface proteins of *B. burgdorferi* to find *Borrelia*-specific epitopes. Through this approach, we successfully identified several linear IgM epitopes, including sequences derived from FlgE, the 41 kDa flagellin core, and the flagellar hook-associated protein FliD, also known as Hook-associated protein 2.

Notably, identifying epitopes in conserved regions of these proteins underscores their potential to elicit a robust and specific antibody response. This finding highlights their relevance as potential targets for neutralizing antibodies, as conserved regions are less likely to undergo antigenic variation, enhancing diagnostic and therapeutic utility [50,51,52,53,54,55]. These results provide a strong foundation for further exploration of these epitopes in developing diagnostic assays or immune-based interventions for *Borrelia* infections.

The Bburg/12/huM epitope was localized in the α1 region, while Bburg/13/huM and Bburg/14/huM were found at the N-terminus and C-terminal regions, respectively, both of which are considered conserved regions [52,53]. Previous research, which identified the N-terminal region (residues 16–27) of OspA as a structurally flexible tethering domain comprising only 12 residues [56,57], provided information regarding the location of the final exposed epitope on the surface.

Synthetic peptides have been gaining prominence in diagnostic tests because of their low cost, rapid production (without worrying about cell culture and protein purification), and high sensitivity and specificity [45,58,59].

The FliD protein, also known as flagellar hook-associated protein 2, plays a key role in the assembly and stability of the *B. burgdorferi* flagellum. In this study, the mapping of epitopes recognized by sera from patients with Lyme disease revealed the sequence 8PGLESKYN15 in the N-terminal region of FliD. The C- and N-terminus regions, as well as D1, are highly conserved [60]. The presence of an epitope in this region suggests that, in addition to generating a robust production of specific antibodies in response to infection, it may be a relevant target for neutralizing antibodies. In this study, we identified FlgE epitopes that have great potential as targets for diagnosing Borreliosis in patients [5]. Among them, the epitopes 321GYGMGYME328 and 381VRIGETGLAGLGDIR395 located between the D1 and D2 domains of the FlgE protein stand out as a possible diagnostic marker, given their location in the structure and accessibility. The structure of the *Borrelia* FlgE protein is composed of three main domains: D0, which forms the inner core of the tubular structure, and the D1 and D2 domains, which integrate the outer layers of the structure [59]. The specific intermolecular cross-linking of lysin-alanine (Lal) between the D1 and D2 domains plays a central role in the structural stability of the flagellar protein. The structural relevance of Lal in FlgE was highlighted by the study with *Treponema denticola* FlgE, deficient in Lal cross-linking, which presented normal hook assembly and cell morphology but with impaired motility, as described by Lynch et al. [56]. These findings suggest that Lal cross-linking may play a key role in the structure and flagellar function, highlighting Lal as a possible antimicrobial target to inhibit pathogenic spirochetes. Finally, the detection of the flagellar hook gene (flgE) in blood samples from individuals diagnosed with *Borrelia* infection corroborates the applicability of this epitope in diagnostic strategies.

We also identified two peptides belonging to the putative outer membrane protein BBA03 and two peptides belonging to outer surface protein A. A previous study showed that the localization of these highly immunogenic and diverse proteins on the membrane allows their exposure [52]. The epitopes identified in the BBA03 protein in the coil and α1 region in this study corroborate the understanding of the functional role of this protein in *B. burgdorferi* [57]. Its conformation presents seven α-helices and regions conserved among different genera of *Borrelia*, which may confer protection against multiple strains. The function of BBA03 becomes even more prominent in mixed infections, where *B. burgdorferi* coexists with other spirochetes [57], suggesting its importance in competitive environments.

Epitopes ^56^ATVDKLELKGTSDKN^70^ and ^266^EGSAVEITKL^275^ mapped in our study correspond with previously identified antibody binding. The first epitope is found in the conserved region, mainly among the six OspA serotypes [61,62]. Of the two identified, EGSAVEITKL is found in a poorly conserved region [51,52]. OspA has been widely studied as an immunogen, only providing 76–92% protection [57,58,59,60,61,62,63]. Although it is a surface protein, it has variations among different serotypes of *B. burgdorferi*; the epitopes may vary depending on the prevalent serotype in the region studied [64,65].

Synthetic peptides have been gaining prominence in diagnostic tests because of their low cost, rapid production (without worrying about cell culture and protein purification), and high sensitivity and specificity [66,67,68]. Therefore, to assess the performance of the mapped epitopes, we synthesized four *Borrelia* epitopes and evaluated their immunoreactivity in anti-IgM testing using ELISA.

The peptide Bburg/01/huM demonstrated a sensitivity of 82.4% and specificity of 56.2%, whereas the peptide Bburg/13/huM had a sensitivity of 47.37% and specificity of 87.18%. These two peptides showed antagonistic profiles, indicating a limited ability to identify infected individuals but a remarkable capacity to identify non-infected individuals. The latter result is consistent with studies that showed low sensitivity for early acute infections [69,70]. In contrast, peptides Bburg/02/huM and Bburg/06/huM demonstrated a high sensitivity of ≥94.74% and specificity of ≥97.44% (Table 2).

The existence of different serotypes could generate a variation in the reactivity of the patient’s serum, thus influencing the identification of epitopes. However, a BLAST performed with the main pathogenic strains showed that the selected peptides are genus-specific.

Though commercial MTTT tests show consistency comparable to the standard method, previous studies showed that the combination of selected epitopes can result in higher sensitivity (76.0%) and specificity (97.4%) [71,72]. Therefore, combining single epitopes is expected to result in tests with better sensitivity and specificity [73]. In fact, recently, a point-of-care serological assay for LBD leveraging IgM-specific synthetic peptides derived from different proteins and used in a paper-based platform for rapid and cost-effective diagnosis has been developed [50].

We produced bi-specific peptides, but the optical density obtained was inferior to the single peptides. In our study, the ELISA results using the chimeritope revealed a sensitivity of 85.0%, specificity of 100.0%, and *p*-value of 0.001. However, previous studies have revealed cross-reactivity between serological tests for syphilis [74], viral infections [75], and IgG anti-*Borrelia* antibodies in *Yersinia*. Given the above, we proceeded with an assay to evaluate polypeptides specific to IgM antibodies in borreliosis using serum samples obtained from individuals diagnosed with leptospirosis and syphilis. The results revealed a substantial difference between the disease groups, as evidenced by the median and *p*-values.

Although *B*. *burgdorferi* antigens contain private antigenic determinants, studies have shown that some common epitopes can cross-react with other pathogens and antibodies associated with autoimmune diseases [68,76]. To address this, we analyzed the seroreactivity of populations with a high prevalence of these co-occurring infections, such as leptospirosis and syphilis patients, to the *B. burgdorferi* bi-epitope antigen. The results revealed significant differences between the disease groups, as evidenced by media and *p*-values. Therefore, the findings suggest that the chimeritope antigen can effectively distinguish between sera from patients with borreliosis and leptospirosis, syphilis, and healthy individuals (Figure 4). However, a low immune response was observed in some patients. This highlights the potential influence of genetic factors or time of sera collection on serological reactivity. This hypothesis warrants further investigation in larger cohorts to better understand the variability in immune responses and refine diagnostic accuracy.

The results demonstrated strong antigenicity of the identified linear IgM epitopes and their specificity, as confirmed through cross-reactivity testing with sera from individuals with heterologous infections. Additionally, in silico analysis using the Protein Information Resource (PIR) database supported the genus-specific nature of the selected peptides. Despite these robust validations, a key limitation of this study was the lack of experimental evaluation across diverse pathogenic Borrelia genospecies. Notably, genospecies, such as *B. garinii* and *B. afzelii*, prevalent in Europe and Asia, were not included in the in vitro assays [20,77]. Future studies should incorporate these variants into multi-center panels comprising clinical sera from geographically diverse regions. Broader validation is essential to determine whether the identified epitopes retain diagnostic accuracy across the Borrelia complex.

## 5. Conclusions

By integrating previously established procedures, including Spot-synthesis and BLAST homology searches, this study successfully identified IgM target epitopes with significant potential for use in a specific early diagnostic assay for Borreliosis. The method demonstrated high sensitivity and specificity, underscoring its effectiveness in detecting private key epitopes. The tested bi-epitope polypeptides yielded promising results, notably eliminating false-positive results.

The study highlights the reliability of these epitopes as diagnostic markers for early *Borrelia* infections, attributed to their difference of reactivity with weak significance as leptospirosis (*p* = 0.0424) and their conservation across *Borrelia* genospecies. These findings pave the way for developing enhanced diagnostic tools for *Borrelia* infections, enabling more accurate diagnoses and improved treatment strategies. Moreover, combining multiple peptide epitopes in a chimeric protein or with a machine learning-based diagnostic model further enhances sensitivity without compromising specificity, showcasing a synergistic approach to advanced diagnostic solutions.

## Figures and Tables

**Figure 1 biomedicines-13-01930-f001:**
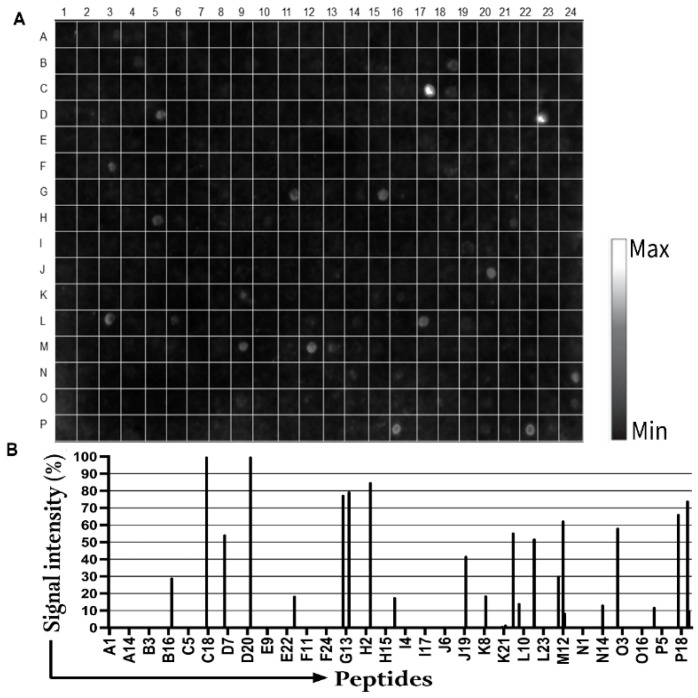
Mapping of linear B-cell epitopes in proteins (Q44767, P11089, O51173, Q44849, and P0CL66) from *B. burgdorferi* using a pool of sera (*n* = 7) from Borreliosis: (**A**) Chemiluminescent assay image showing IgM-reactive peptides. (**B**) Graph depicting the signal intensity of the immunoreactive peptides. The x-axis represents the peptide distribution on the membrane, and the y-axis shows the relative signal intensity in percentage.

**Figure 2 biomedicines-13-01930-f002:**
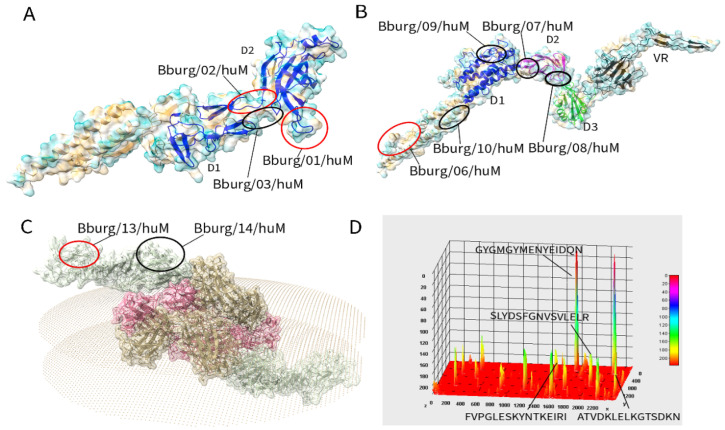
Three-dimensional structural model protein from *Borrelia burgdorferi* and localization of the immunodominant IgM epitopes. The picture represents the location of the epitopes on the ribbon diagram of FlgE (**A**), (Osp A) outer surface protein (**B**), (Flg 41 kDa) flagellar filament protein (**C**), and (Flg hook 2) flagellar protein (**D**). The molecular modeling of the proteins was based on homology using an AlphaFold v2.0 script and crystal structure of OSPA (PDB:1fj1). The fourteen highly immunogenic IgM epitopes mapped are highlighted in blue, and the four epitopes used for the peptide-ELISA assay are depicted.

**Figure 3 biomedicines-13-01930-f003:**
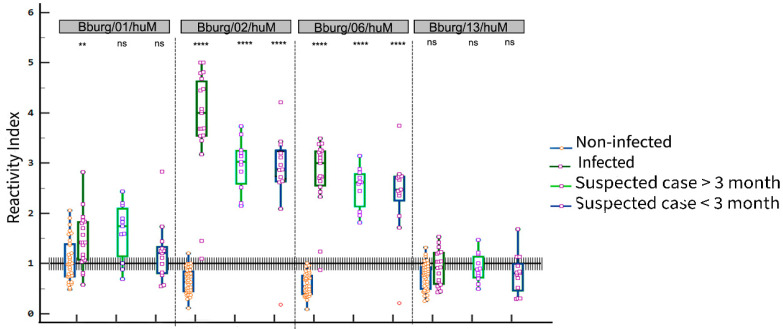
Reactivity Index (RI) of synthetic peptide reactive in individuals with anti-Borrelia IgM (*n* = 19), non-infected (*n* = 39), infected SC < 3 months (*n* = 13), SC > 3 months (*n* = 11). The dashed line represents the reactivity index (RI) cutoff value. The area delimited by a gray rectangle indicates the indeterminate zone (RI ± 10%). Significance level *p*: (ns) *p* > 0.05; (**) *p* ≤ 0.01; (****) *p* ≤ 0.01; SC, Suspected case.

**Figure 4 biomedicines-13-01930-f004:**
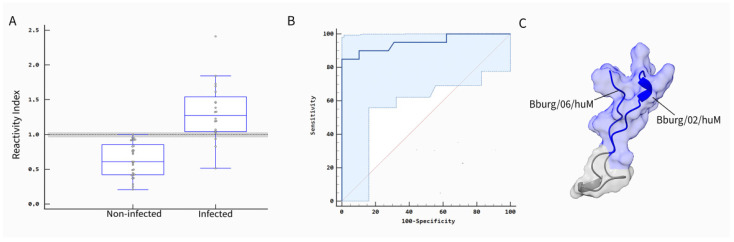
Performance parameters of the bi-specific peptides for diagnostic confirmation. (**A**) Reactivity of the peptide (0.5 µg /well) with IgM in ELISA using seronegative serum sample (*n* = 29) and seropositive serum sample (*n* = 20). The dashed line represents the gray zone, defined as positive (>1.1), indeterminate (1.1–09), or negative (<0.9). (**B**) ROC (receiver operating characteristic) curve for the ELISA assay indicating the cutoff, sensitivity (85.0), specificity (100.0), and AUC (area under the curve; 0.826). (**C**) Structural prediction of the bi-specific peptide. *p* ≤ 0.001.

**Figure 5 biomedicines-13-01930-f005:**
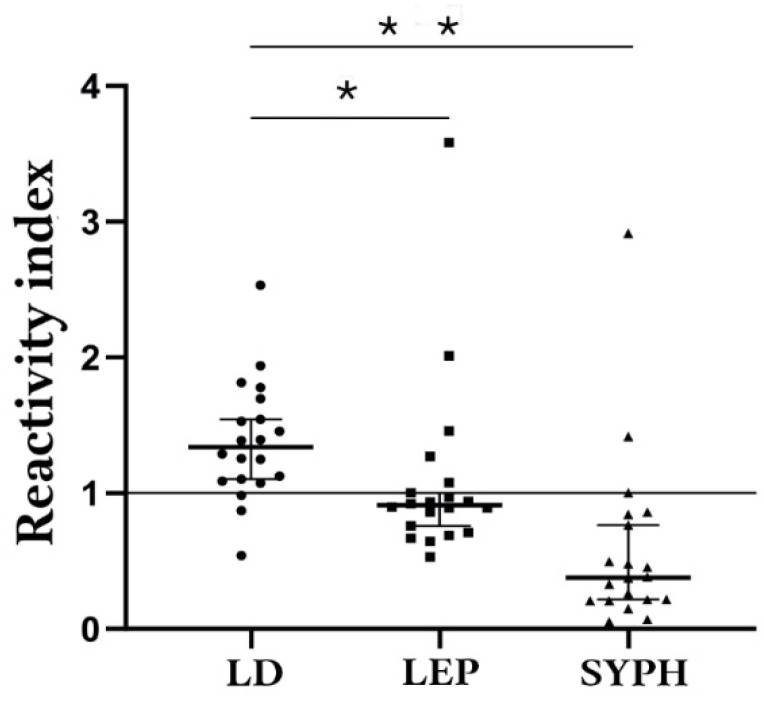
Box and whisker plot comparing ELISA-polypeptide IgM cross-reactivity using sera samples from borreliosis (●; LD, *n* = 20), leptospirosis (■; LEP, *n* = 20), and syphilis (▲; SYPH, *n* = 20) individuals. The Kruskal–Wallis test result graph displays the *p*-value (*p* < 0.0001 (**); *p* = 0.0424 (*)).

**Table 1 biomedicines-13-01930-t001:** Clinical characteristics of patients.

	Seropositive Cases	Suspected Cases
Clinical signs	Erythema migrans, arthritis, neurological abnormalities, cardiac involvement, relapsing symptoms. Chronic fatigue or cognitive disturbances.	Erythema migrans, arthritis, neurological abnormalities, cardiac involvement, relapsing symptoms. Chronic fatigue or cognitive disturbances.
Serological status	Positive	Negative
Epidemiological exposure	History of tick bite and/or environmental exposure in endemic areas.	History of tick bite and/or environmental exposure in endemic areas.

**Table 2 biomedicines-13-01930-t002:** List of B-cell linear IgM epitopes with signal intensity ≥ 30%, peptide match, secondary structure (C, coil; H, helix; S, strand) predicted by I-TASSER (accessed on 20 November 2022).

Protein	Epitope	2nd Structure *	a Number	Peptide Match **
FlgE	Bburg/01/huM	C	209 SLYDSFGN 216	Various *Borrelia* sp.
	Bburg/02/huM	C	321 GYGMGYME 328	Various *Borrelia* sp.
	Bburg/03/huM	C+S	381 VRIGETGLAGLGDIR 395	Another organism
Flg41 kDa	Bburg/04/huM	H+C	121 ANLSKTQEKLSSGYR 135	Another organism
	Bburg/05/huM	H+C	322 AQANQVPQYVLSLLR 336	Another organism.
Flg hook2	Bburg/06/huM	C	08 PGLESKYN 15	Various *Borrelia* sp.
	Bburg/07/huM	C+S	76 SGNSSNSEVLTLSTR 90	Another organism
	Bburg/08/huM	C+S	391 AENAKIKFDGVDVER 405	Another organism
	Bburg/09/huM	H+C+S	546 RYLRLDEKKFDESIR 560	Another organism
	Bburg/10/huM	H	616 QKNKVEDYKKKYEDR 630	Another organism
BBA03	Bburg/11/huM	C	31 DEKSQAKSNLVD 42	Another organism
	Bburg/12/huM	H+C	46 IEFSKATPLEKLVSR 60	Another organism
OSP A	Bburg/13/huM	C	56 ATVDKLELKGTSDKN 70	Various *Borrelia* sp.
	Bburg/14/huM	C	266 EGSAVEITKL 275	Another organism

C, coil; H, helix; S, strand; * based on an AlphaFold analysis; ** PIR (Protein Information Resource).

**Table 3 biomedicines-13-01930-t003:** The diagnostic assessment of the ELISA-peptide and Receiver operating characteristic curve (ROC) with 95% confidence intervals.

						Suspected Case				
	Infected					>3 Months		<3 Months		
Peptide	Se (%)	Sp (%)	AUC	Ac (%)	Se (%)	Sp (%)	AUC	Se (%)	Sp (%)	AUC
Bburg/01/huM	82.4	56.2	0.721	66.6	72.7	56.2	0.557	76.9	59.4	0.671
Bburg/02/huM	100.0	97.44	0.999	96.2	100.0	100.0	1	92.3	100.0	0.925
Bburg/06/huM	94.74	100.0	0.996	98.2	100.0	100.0	1	92.3	100.0	0.925
Bburg/13/huM	47.37	87.18	0.652	74.1	72.7	59.0	0.622	76.9	59.4	0.694

The area under the curve (AUC), Accuracy (Ac), sensitivity (Se), and specificity (Sp).

**Table 4 biomedicines-13-01930-t004:** Kruskal–Wallis test analysis of the reactivity of the *Borrelia* polypeptide with sera from leptospirosis and syphilis individuals.

Comparison Group	Median	75% Percentile	25% Percentile
Lyme-Borreliosis	0.2613	0.3235	0.2131
Leptospirosis	0.0755	0.08788	0.060
Syphilis	0.02175	0.04738	0.01213

## Data Availability

The data presented in this study are available on request from the corresponding author.

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
