# Peer review of "Lyme-Borreliosis Disease: IgM Epitope Mapping and Evaluation of a Serological Assay Based on Immunodominant Bi-Specific Peptides"

_biomedicines, 2025, doi:10.3390/biomedicines13081930_

Round 1
Reviewer 1 Report
Comments and Suggestions for Authors
- In abstract, the authors used some abbreviations before describing full words such as SPOT, ELISA etc. Please revise at the revised manuscript.
- Regarding the informed consent statement, the authors described the institutional review board statement only. Please add the authors took written informed consent or not under the materials and methods section.
- The readers understood that ROC analysis was done based on the different concentration of antigen , At Figure-4, ROC represents for which concentration of antigen, Please add the specific ROC curve for antigen at the revised manuscript.
- Please clear describe that the number of seropositive case and the number of suspected cases separately.
- The authors used ELISA as the gold standard test in this study. Moreover, the authors also stated that the ELISA test also showed cross reactivity with syphillis antibody? Is it possible to use ELISA as gold standard to validate your assay?
- Moreover, the readers would like to know ho many suspected cases could be confined as laboratory confirmed cases in this study? Please clearly describe at the revised manuscript.
Author Response
Reviewer 1
- In abstract, the authors used some abbreviations before describing full words such as SPOT, ELISA etc. Please revise at the revised manuscript.
R: SPOT synthesis is a method that employs standard solid-phase peptide synthesis (SPPS) on a cellulose membrane (1,2) and is not an abbreviation. Furthermore, I have made the necessary revisions in the Methods section (lines 7–10).
- The authors described the institutional review board statement only regarding the informed consent statement. Please add whether the authors obtained written informed consent under the materials and methods section.
R: We appreciate the reviewer’s comment. The serum samples used in this study were obtained from external laboratories, specifically the Rheumatology Division at the University of São Paulo, School of Medicine (USP), the Laboratory of Biodiversity in Entomology at Instituto Oswaldo Cruz (IOC/FIOCRUZ), and HEMORIO. These institutions collected the samples under their respective ethical approvals, which included obtaining informed consent from participants. This information has been clarified in the Materials and Methods section.
- The readers understood that ROC analysis was done based on the different concentrations of antigens. In Figure-4, ROC represents which concentration of antigen. Please add the specific ROC curve for the antigen to the revised manuscript.
R: We thank the reviewer for the valuable comment. The ROC analysis shown in Figure 4 was performed using a single standardized antigen concentration of 0.5 µg/well, applied equally to both the individual peptides and the bi-specific peptide, as described in the Methods section.
To prevent any misunderstanding regarding the use of multiple concentrations, we have revised the manuscript text and explicitly stated in the legend of Figure 4 that the ROC curve refers specifically to this single antigen concentration.
- Please clearly describe the number of seropositive cases and the number of suspected cases separately.
R: The revised manuscript now clearly describes the number of seropositive cases and the number of suspected cases separately. This distinction has been added in the Results section and presented in Table 1 to improve clarity.
- The authors used ELISA as the gold standard test in this study. Moreover, the authors stated that the ELISA test also showed cross-reactivity with the syphilis antibody. Is it possible to use ELISA as the gold standard to validate your assay?
R: In this study, ELISA was not employed as a gold standard diagnostic test but as an analytical tool to assess the immunoreactivity of synthetic peptides with anti-Borrelia burgdorferi IgM antibodies. We acknowledge that conventional ELISA assays may exhibit cross-reactivity with antibodies against Treponema pallidum, as reported in the literature. However, our aim was to evaluate the diagnostic potential of the selected peptides under controlled conditions. Future studies will include comparative analyses with another serological assay to provide a more comprehensive validation and to address the potential impact of cross-reactivity.
- Moreover, the readers would like to know how many suspected cases could be confined as laboratory-confirmed cases in this study? Please clearly describe at the revised manuscript
R: We have now clearly specified the number of suspected cases that were confirmed as laboratory-confirmed cases based on peptide-based ELISA results. Among the 24 suspected cases, 23 were confirmed based on a Reactivity Index (RI) > 1.0 in at least one of the tested peptides. This information has been included in the Results section.

Reviewer 2 Report
Comments and Suggestions for Authors
The manuscript titled “Lyme-Borreliosis Disease: IgM Epitope Mapping and Evaluation of a Serological Assay Based on Immunodominant Bi-Specific Peptides” " by Mônica E. T. A. Chino et. al. to identified ten IgM epitopes,the ELISA-peptide assay demonstrated a sensitivity of ≥ 3087.3%, specificity of ≥ 56.2%, and accuracy of ≥ 66.6%. It's a very interesting thing to do.Below are certain points that need further attention.
- IgM ELISA-peptide assay to detect sensitivity, whether the authors considered comparisons with existing methods.
- How did the author determine ELISA cut-off.
- With respect to specificity, it was not observed whether other pathogens were used
- The citation is not standard enough for example [58] [49–53].
Author Response
The manuscript titled “Lyme-Borreliosis Disease: IgM Epitope Mapping and Evaluation of a Serological Assay Based on Immunodominant Bi-Specific Peptides” " by Mônica E. T. A. Chino et. al. to identified ten IgM epitopes, the ELISA-peptide assay demonstrated a sensitivity of ≥ 3087.3%, specificity of ≥ 56.2%, and accuracy of ≥ 66.6%. It's a very interesting thing to do. Below are certain points that need further attention.
- IgM ELISA-peptide assay is used to detect sensitivity and whether the authors considered comparisons with existing methods.
R: Thank you for your insightful comment. While we did use the ELISA method in this study, the primary focus was on evaluating the synthetic peptide's performance in its reactivity with IgM antibodies. Our goal was not to directly compare this assay with standard ELISA methods for detecting Lyme disease. Future studies could include comparisons with established diagnostic methods to further validate the utility of this assay in clinical settings.
- How did the author determine ELISA cut-off.
R: The ELISA cut-off (CO) value was determined based on the ROC curve analysis, as described in the Methods section. The CO was set at the point that maximized both sensitivity and specificity, ensuring the optimal balance between true positive and false positive rates.
- With respect to specificity, it was not observed whether other pathogens were used
R: In the present study, cross-reactivity was indeed evaluated using serum samples from patients with syphilis (n = 20) and leptospirosis (n = 20), in addition to samples from Lyme borreliosis (LB) patients (n = 20). Although a few cross-reactive samples were detected, the significantly higher reactivity in the LB group underscores the specificity of the bi-peptide-based assay for Lyme borreliosis. The revised manuscript clearly highlights these findings (Figure 5 and Table 3).
- For example, the citation is not standard enough [58] [49–53].
R: We have carefully reviewed and corrected the formatting of all in-text citations to ensure they follow the journal’s standard reference style.

Reviewer 3 Report
Comments and Suggestions for Authors
Overall, this study is interesting and shows several significant findings. I just have several minor suggestions.
- Please remove the first sentence of the introduction.
- The introduction is too lengthy. Please shorten it.
- May add one table to summarize the clinical characteristics of included patients in this study.
Author Response
Overall, this study is interesting and shows several significant findings. I just have several minor suggestions.
- Please remove the first sentence of the Introduction.
R: Thank you for your suggestion. As requested, we have removed the first sentence of the Introduction in the revised manuscript.
- The Introduction is too lengthy. Please shorten it.
R: Thank you for your feedback. We have revised and shortened the Introduction section to improve focus and readability. Redundant or overly detailed content was removed to ensure the section remains concise while retaining the necessary scientific background and rationale for the study.
- One table may be added to summarize the clinical characteristics of the patients included in this study.
R: We appreciate the reviewer’s valuable suggestion. We added a new table (Table 1) that summarizes the clinical characteristics of the patients in this study, including diagnostic category, clinical signs, serological status, and epidemiological exposure. This addition aims to improve clarity and provide a concise overview of the study population.
Reference
- Winkler DFH. SPOT Synthesis: The Solid-Phase Peptide Synthesis on Planar Surfaces. Methods Mol Biol. 2020;2103:151-173. doi: 10.1007/978-1-0716-0227-0_10. PMID: 31879924.
- De-Simone SG, Napoleão-Pêgo P, Lechuga GC, Carvalho JPRS, Monteiro ME, Morel CM, Provance DW Jr. Mapping IgA Epitope and Cross-Reactivity between Severe Acute Respiratory Syndrome-Associated Coronavirus 2 and DENV. Vaccines (Basel). 2023 Nov 24;11(12):1749. doi: 10.3390/vaccines11121749.

Reviewer 4 Report
Comments and Suggestions for Authors
The manuscript by Mônica E.T.A. Chino and colleagues provides a well-executed investigation into the antigenic landscape of Borrelia burgdorferi, focusing on linear IgM epitopes derived from five surface proteins. Using SPOT synthesis, ELISA, and ROC analysis, the authors identified several conserved regions, notably in FlgE, FliD, OspA, and BBA03, that hold promise for serodiagnostic development. The work is technically sound, well-organized, and relevant to the ongoing need for improved diagnostics in early Lyme borreliosis.
Some minor comments and suggestions are given below.
Introduction section:
- Line 49-50. "...different parties in Mundo....". Please clarify. Replace “Mundo” with appropriate English equivalent.
- Line 64-65: "...therapeutic wandering...". Please clarify. Maybe using "...misdiagnosis and inappropriate treatment..." sould be better.
- Maybe Authors could justify/explore why identifying new IgM epitopes is a better strategy compared to modifying current tools.
Material and Methods
- Clarify the total number of positive/suspected cases — currently, n=11 and n=13 add up to 24, not 21. Likely a typo or overlap; revise.
- Line 199: Instead of “plates were treated,” say “plates were incubated with” for consistency in tone.
Discussion
While the peptides were described as genus-specific based on BLAST analysis, the paper could benefit from a deeper discussion on the performance across different Borrelia genospecies (e.g., B. garinii, B. afzelii, prevalent in Europe), and validation in multi-strain panels or global cohorts. This would strengthen the generalizability of the findings and the practical utility of the identified epitopes.
